# Simulation of Pressure-Driven and Channel-Based Microfluidics on Different Abstract Levels: A Case Study

**DOI:** 10.3390/s22145392

**Published:** 2022-07-19

**Authors:** Michel Takken, Robert Wille

**Affiliations:** 1Department of Electrical and Computer Engineering, Technical University of Munich, Arcisstraße 21, 80333 München, Germany; 2Software Competence Center Hagenberg GmbH (SCCH), Softwarepark 32a, 4232 Hagenberg, Austria

**Keywords:** microfluidic, Labs-on-a-Chip, 1D simulations, finite volume method, *Lattice-Boltzmann Method*

## Abstract

A microfluidic device, or a Lab-on-a-Chip (LoC), performs lab operations on the microscale through the manipulation of fluids. The design and fabrication of such devices usually is a tedious process, and auxiliary tools, such as simulators, can alleviate the necessary effort for the design process. Simulations of fluids exist in various forms and can be categorized according to how well they represent the underlying physics, into so-called abstraction levels. In this work, we consider simulation approaches in 1D, which are based on analytical solutions of simplified problems, and approaches in 2D and 3D, for which we use two different *Computational Fluid Dynamics* (CFD) methods—namely, the *Finite Volume Method* (FVM) and the *Lattice-Boltzmann Method* (LBM). All these methods come with their pros and cons with respect to accuracy and required compute time, but unfortunately, most designers and researchers are not aware of the trade-off that can be made within the broad spectrum of available simulation approaches for microfluidics and end up choosing a simulation approach arbitrarily. We provide an overview of different simulation approaches as well as a case study of their performance to aid designers and researchers in their choice. To this end, we consider three representative use cases of pressure-driven and channel-based microfluidic devices (namely the non-Newtonian flow in a channel, the mixing of two fluids in a channel, and the behavior of droplets in channels). The considerations and evaluations raise the awareness and provide several insights for what simulation approaches can be utilized today when designing corresponding devices (and for what they cannot be utilized yet).

## 1. Introduction

A microfluidic chip, or a *Lab-on-a-Chip* (LoC), is a device that performs lab operations on a microscale through a set of fluid manipulations. A prominent and established kind of LoCs are pressure-driven and channel-based devices [1] in which fluids are flowing through a configuration of channels with a scale in the order of micrometers or, in some cases, even nanometers (note that alternative LoCs, e.g., based on paper-based microfluidics [2] or *electrowetting on dielectric* (EWOD, [3]) exist but are not considered in this work). The advantage of working on this scale is that less chemical components are required and chemical reactions usually take less time [4]. However, the design and fabrication of corresponding devices is still a time-consuming and tedious process: channels must be appropriately dimensioned, pressure and flow rates must be accordingly adjusted, and even the slightest changes may completely change the behavior.

Simulations are an essential tool in this process as they can offer an early evaluation of the behavior of a design—even in the absence of a fabricated prototype. To this end, a huge variety of complementary simulation approaches have been introduced in the past [5,6,7]. They operate on different abstraction levels, i.e., simplifications of the real-world physics to model the respectively needed physical phenomena. Figure 1 provides a rough overview of simulation approaches on the abstraction-cost scale. Examples of approaches on a high abstraction level are 1D approaches as introduced and used, e.g., in [5,8]. These approaches are usually based on analytical solutions of simplified problems and the computational effort scales linearly with the mesh refinement (i.e., the computational complexity of these approaches is O(n), where *n* is the amount of data points along one dimension of the problem geometry). Approaches on a lower abstraction level rely on *Computational Fluid Dynamics*(CFD, [9]) and, more precisely, on methods such as the *Finite Volume Method* (FVM, [9,10]), and the *Lattice-Boltzmann Method* (LBM, [11,12]). These can be further divided into 2D and 3D approaches, i.e., into methods considering a 2-dimensional and a 3-dimensional space, and, hence, scale in a quadratic ($O(n^{2})$) and cubic ($O(n^{3})$) fashion, respectively. Evaluations and case studies such as conducted in [8] illustrate the potential and showcased how the use of simulations can reduce the design time of an LoC substantially (in some of those studies, design time reductions from an entire month to a single day have been observed).

However, simulations are not perfect and, although helpful, do not necessarily reflect the real-world behavior (i.e., the ground truth). In fact, depending on the considered case, the used simulation approach, the made assumptions and configurations, etc., different behavior of a given design may be concluded from a simulation run. The FVM approach alone can give different simulation results, depending on, e.g., the spatial and temporal discretization, the quality of the mesh, or even the obtained order of spatial accuracy for higher-order schemes [10]. A review done in [13] provides a broad overview of possible methods within the FVM approach to model the interface of a multiphase flow only; and a case study done in [14] showcases that, for a single simulation approach (FVM), even the chosen tool can significantly influence the required computational time and simulated results. The same can be said for other CFD methods, such as the LBM. Despite this broad spectrum of available simulation approaches for microfluidics, most designers and researchers often only adopt a single simulation approach for their microfluidic simulations [15,16,17,18,19,20]. Moreover, conducting these simulations may be computationally expensive—with simulations on lower abstraction levels easily taking days or even weeks to complete [14] while, in contrast, simulations on higher abstraction levels might be faster but may not provide the desired accuracy. Not knowing the spectrum and capabilities of the single simulation methods at the different abstractions levels makes it hard, if not impossible, to properly trade-off the desired quality of the possible results against the computational effort needed to generate them.

In this work, we aim at shedding light into this by providing (1) an overview of different and complementary simulation methods as well as (2) a case study of their performance for microfluidic designs (note that the scope of this work is to raise the awareness and discuss the potential of simulation and, hence, does not include experimental data obtained from physically produced microfluidic devices). To this end, we consider three typical and representative use cases of pressure-driven and channel-based microfluidic devices (namely the non-Newtonian flow in a channel, the mixing of two fluids in a channel, and the behavior of droplets in channels; covered in more detail in Section 2). For each of those use cases, its corresponding representation according to the 1D, 2D, and 3D abstraction level is considered. Based on that, we review representative simulation methods (covered in Section 3) and describe how the considered use cases can be properly simulated with them (covered in Section 4). Results of those simulations (covered in Section 5) are discussed and showcase the benefits, but also the pitfalls, of simulations at different abstraction levels.

By this, this case study certainly does not claim to comprehensively cover the problem or to allow for generic conclusions about what simulation method and abstraction level performs best for what use case. However, the reviews, results, and discussions provided in this paper certainly raise awareness of the spectrum of available solutions and its applicability with respect to computational efforts vs. quality. More precisely, this work provides (besides others) the following main insights:Simulations at high abstraction levels come with less computational costs than simulations at lower abstraction levels. While not surprising at all, this is, to the best of our knowledge, the first case study that investigates the performance of simulations and quantifies the computational cost for microfluidic devices at different abstraction levels.Applying more computational efforts does not always yield a better simulation quality. In fact, we show that for certain use cases, applying a (computationally more expensive) lower abstraction does not significantly improve the accuracy of the results. Insights like those enable the designers or researcher to properly trade-off the available approaches and to save a lot of simulation time without a severe less of quality.The choice of the applied simulator should reflect the need of the designer or researcher. If, e.g., droplets are considered and the end-user is only interested in the position of them, the 1D abstraction level might be sufficient (providing a very fast solution). If, instead, also the integrity or the deformation of the droplet is of interest, the more accurate but also much slower 2D/3D approach might be needed. Again, insights which equip the designers or researcher to choose the right solution for the task.For some use cases, simulations provide inconclusive results. This shows that, although simulations really can help in many use cases, they might not always be the “ground-truth”. After all, they *do not* reflect the real world in a perfect fashion but heavily rely on the underlying models and assumptions. Being aware of that and the complementary approaches helps designers and researchers to constantly reflect when a certain simulation can be trusted and, in case of contradicting results, which simulation approach most likely covered the real world best.

Overall, this case study certainly does not aim to establish a ground truth for simulation of microfluidic devices (after all, the fabrication and evaluation of actual prototypes are still needed for that). However, it offers an overview and several insights for what simulation approaches can be utilized today when designing corresponding devices (and for what they cannot be utilized yet).

## 2. Considered Use Cases

The goal of this work is to provide an overview and evaluation of the performance of different and complementary approaches for the simulation of fluidic behavior in pressure-driven channel-based microfluidic devices. To this end, a representative set of use cases (covering different phenomena in fluid physics) is key. In this section, we introduce the use cases that have been considered in our case study, namely non-Newtonian fluid flow, fluid mixing, and droplet microfluidics, as well as motivate their importance in the field of microfluidics. All of them rely on the flow of a fluid through a rectangular channel which is reviewed first. Afterwards, each use case is described.

### 2.1. Fluid Flow through a Channel

The flow of a fluid through a channel is obviously the basis for any simulation of a channel-based microfluidic device. The corresponding physics can be analytically described through the Hagen–Poiseuille law defined by
(1)$$Q=\frac{\pi r^{4}}{8\mu l}\Delta p,$$
where *r* is the radius and *l* is the length of the channel, Δp is the applied pressure difference, μ is the viscosity of the fluid, and *Q* describes the mass flow rate. However, this law can only be applied if *cylindrical* channels are considered; while, in microfluidics, channels usually have a *rectangular* shape. Accordingly, the Hagen–Poiseuille law needs to be modified. More precisely, for a rectangular channel with width *w* and height *h*, the mass flow rate is given by [21]
(2)$$Q=\frac{wh^{3}\Delta p}{12\mu l}\left[1-\frac{h}{w}\left(\frac{192}{\pi^{5}}\sum_{n=1,3,5}^{\infty }\frac{1}{n^{5}}tanh\left(\frac{n\pi w}{2h}\right)\right)\right].$$Here, the summation can be truncated after the second term, with a truncation error of $O(10^{-4})$. The mass flow rate can be used to get the average flow velocity of the fluid. However, inside a channel, the flow velocity is not equal everywhere. The fluid is stagnant at the walls, and the largest fluid velocity is found furthest away from the channel walls, in the middle of the channel. This is due to the shear stress, which occurs because molecules stick to each other and pull each other; creating horizontal layers of fluid flowing at different speeds, in a laminar fashion. The viscosity of a fluid describes the resistance to the rate of deformation. The distribution of flow velocity along the cross-section of a channel is described by the flow profile.

**Example** **1.**
*For a Newtonian fluid (a fluid in which the viscosity of the fluid depends linearly on the shear strain rate, e.g., rate of deformation) such as water, the flow profile is parabolic. In Figure 2, this is shown by the solid line. More precisely, Figure 2 shows the flow in a channel flowing from the left to the right, with a channel length l and a channel width w, as well as walls on the top and bottom. The arrows on the left indicate the inflow velocity at the inlet, following the Newtonian flow profile.*


Based on the considerations so far, we can analyze flows for channel-based microfluidic devices. However, this only covers the main basic case. As soon as further issues beyond a simple channel flow are considered (such as the use cases considered in this work), purely relying on Equation (Equation 2) is insufficient. An additional analysis is required which is covered next.

### 2.2. Non-Newtonian Fluid Flow

Thus far, we assumed a fluid flow behaving in a Newtonian fashion, i.e., the fluid viscosity linearly depends on the shear strain rate. The shear strain rate of a fluid describes its shear deformation, resulting from external forces, with respect to time. A fluid’s viscosity can thus be said to be a measure of its resistance against deformation. For real world problems, the corresponding fluids cannot always be assumed to behave in a Newtonian fashion. In particular in applications, e.g., from the food industry, material science, and in the medical field, fluids frequently show a nonlinear relation between the viscosity and the rate of deformation and, hence, show non-Newtonian properties [22]. Accordingly, this constitutes a good representative case for the evaluations conducted in this work.

**Example** **2.**
*Examples of non-Newtonian fluids are shear-thinning or shear-thickening fluids, i.e., the fluid viscosity decreases or increases with increased shear strain rate, respectively. The dashed lines in Figure 2 indicate the corresponding flow profiles for shear-thinning and shear-thickening fluids. Here, it is obvious how for, e.g., shear-thinning fluids, the rate of deformation of the fluid near the wall is significantly high; causing the flow to have a more uniform flow speed in the center region of the flow channel. Shear-thickening fluids will get a sharper flow profile, due to the higher resistance against deformation of the fluid.*


The flow profile for non-Newtonian fluids can be described by using rheological models, such as the power-law [23], the Casson model [24], and the Carreau–Yasuda model [25,26]. In this case study, we will model the shear-thinning behavior of blood in a rectangular channel, using the Carreau–Yasuda model. This model gives the effective viscosity $\mu_{\mathrm{eff}}$ as a function of shear strain rate $\dot{\gamma }$ as
(3)$$\mu_{\mathrm{eff}}\left(\dot{\gamma }\right)=\mu_{\infty }+\left(\mu_{0}-\mu_{\infty }\right){\left(1+{\left(\lambda \dot{\gamma }\right)}^{a}\right)}^{\frac{n-1}{a}}.$$Here, $\mu_{0}$ and $\mu_{\infty }$ are the viscosity at zero and infinite strain rate, respectively, λ is the relaxation time, and *n* and *a* are power indices. These parameters are fluid-dependent and we will use the following values for blood (which are in line with experimental data [27]):$$\begin{array}{cc} \mu_{0} & =22e^{-3}Pa\cdot \;s\\ \mu_{\infty } & =2.2e^{-3}Pa\cdot \;s\\ a & =0.644\\ n & =0.392\\ \lambda & =0.110\;s \end{array}$$

Because microfluidic devices only need a very small amount of specimen to perform tests, compared to regular labs, it is attractive to apply them in the medical field, where specimen fluid is often limited. This use case was chosen due to its relevance for the application of LoCs in the medical field, but it can easily be adapted to other non-Newtonian fluid problems by changing the fluid parameters or the problem domain accordingly.

### 2.3. Fluid Mixing

In this use case, we address the mixing properties of two miscible fluids. Mixing of two or more miscible fluids is a common operation in (bio-)chemistry, and in microfluidics it can be used to, e.g., generate a stream of liquid with predefined concentrations [28]. Mixing is usually induced through stirring, and fluids predominantly mix through convection. For LoCs, however, this is not the case. Due to the laminar nature of flow in microfluidic channels, the mixing operation of two or more fluids depends on the diffusive properties of the fluids, rather than convective transport properties.

The diffusion distance *d* of molecules depends on the diffusion coefficient $D_{D}$ and flow time *t*, and is given by [29]
(4)$$d\approx \sqrt{2D_{D}t}.$$Diffusion is a very slow transport mechanism, and even in microchannels, it can take up to several minutes to achieve full mixing, which requires relatively long flow channels [29]. Mixing can be induced through active mixing methods, such as using actuators [30], peristaltic pumps [31], or electro-kinetics [32]. In this case study, however, we will focus on the passive mixing properties of two fluids in a meandering channel. Meandering channels provide enough mixing time for fluids flowing at a certain speed, whilst not taking up too much area on a microfluidic chip.

**Example** **3.**
*Figure 3 shows an example of passive mixing in a meandering channel (this Figure was authorized for reprinting). On the left, we have two inlets containing fluids of different colors. The mixing mechanism within the meandering channel is illustrated here. It can be seen that the laminarity of the flow causes very little mixing between the two colored streams, and a lot of LoC area is required to accommodate, e.g., meandering channels. The dimensioning of such channels highly depends on the mixing properties of the fluids.*


The mixing of fluids with different concentrations or content is a fundamental (bio-)chemical lab operation. Therefore, reliable and fast predictions of mixing properties are of utmost importance for LoC design, making this use case relevant for our case study.

### 2.4. Droplet Microfluidics

Finally, we consider the droplet microfluidics use case in which the fluid flowing through the channel acts as a carrier fluid for another injected (immiscible) fluid; leading to the formation of droplets which are transported by the carrier fluid. This leads to a two-phase microfluidic system. A network of channels allows the droplet to take different paths. This system can be used to transport bio-chemical assays to the desired location on the chip.

**Example** **4.**
*An example of how microfluidic droplets can take different paths is illustrated using Figure 4. Here, a colored droplet is shown, flowing in from the left, heading towards a bifurcation (T-junction). At the bifurcation, the droplet can either flow to the top or to the bottom. Provided that the droplet does not split, it will choose the path which constitutes the highest total pressure difference, i.e., the path with the lowest hydraulic resistance.*


Combined with the flow speed, the pressure difference of a channel can be expressed as a resistance. The hydraulic resistance of a channel is given by
(5)$$R_{H}=\frac{\Delta p}{Q}.$$Droplet paths can be predicted relatively easily using the laws described in Section 2.1, the hydraulic resistance of channels, and analogous methods from electrical engineering such as Kirchoff’s law [5].

However, droplet deformations depend on the surface tension, which can be described for droplets using
(6)$$p^{c}-p^{d}=\frac{2\gamma^{cd}}{r},$$
where $p^{c}$ and $p^{d}$ are the pressures of the carrier fluid and droplet, respectively, *r* is the radius of the droplet, and $\gamma^{cd}$ is the surface tension coefficient of the two liquids and depends on the temperature. The prediction of whether a droplet splits or not needs the analysis of the surface tension of the droplet. In this use case, we consider a droplet flowing through a bifurcation, as in Figure 4, by which stresses are applied to the droplet surface to observe whether splitting occurs.

Droplet microfluidics provides the basics for several microfluidics applications such as single cell analysis [22], high throughput PCR [33], and material science [22], and is, hence, a relevant use case.

## 3. Applied Simulation Methods

The use cases given in Section 2 can be simulated on different abstraction levels and with different simulation approaches. In this case study, we considered representative approaches of the so-called 1D analysis model and two representatives of established CFD methods, namely the *Finite Volume Method* (FVM) and the *Lattice-Boltzmann Method* (LBM), for the 2D and 3D approaches. For all these approaches, the basic ideas and how we used them to simulate the use cases introduced above are discussed in this section. The discussions of the methods are kept brief, but references for a more detailed treatment are provided throughout the section.

### 3.1. 1D Models

To simulate the behavior of the three use cases, 1D models based on the Hagen–Poiseuille law for rectangular channels (as reviewed in Section 2) can be used. However, for each use case, additional physics need to be accounted for. More precisely:

#### 3.1.1. Non-Newtonian Fluid Flow

For non-Newtonian fluids, models can be used to describe the behavior of the fluid viscosity with respect to the shear strain rate. For the simulation of blood flow, we have chosen the Carreau–Yasuda model (cf. Equation (Equation 3)). The shear strain rate $\dot{\gamma }$ and viscosity μ are local properties and cannot be solved analytically [34]. Therefore, a semi-analytical approach needs to be employed.

Such a semi-analytical approach is provided by [35] for a flow between parallel plates, which is virtually the same as a rectangular channel in the 2D domain. The flow profile between parallel plates is given by
(7)$$u\left(z\right)=\int_{0}^{z}\dot{\gamma }\left(z\right)dz,$$
where *z* is the perpendicular distance measured from the closest wall, and $u_{\max}$ is at half the channel width. The shear strain rate can be numerically approximated through a root finding method, and is obtained from
(8)$$\dot{\gamma }\left(\mu_{\infty }+\left(\mu_{0}-\mu_{\infty }{\left(1+\lambda^{a}{\dot{\gamma }}^{a}\right)}^{\frac{n-1}{a}}\right)\right)=\frac{z\Delta p}{l}.$$

An approximate solution to Equation (Equation 7) can be obtained using a quadrature rule.

#### 3.1.2. Fluid Mixing

The 1D approach for the fluid mixing focuses on predicting the mixing behavior within a rectangular channel of length *l*, and is based on Fick’s law for diffusion in the y-direction across the channel, and advection in the x-direction along the channel [36], i.e.,
(9)$$\frac{\partial^{2}c}{\partial y^{2}}=Pe\frac{\partial c}{\partial x}.$$Here, *c* is the concentration of a species, *x* is the location along the channel length, *y* is the location across the channel width, and Pe is the Péclet number, which is a non-dimensional number that defines the ratio of advection and diffusion of a fluid flow. The exact solution of Equation (Equation 9) is given by [37] as
(10)$$c\left(x,y\right)=0.5+\frac{1}{\pi }\sum_{n=1}^{\infty }e^{\frac{-\pi^{2}{\left(2n-1\right)}^{2}x}{Pe}}sin\left(\pi (2n-1)y\right)\frac{1-cos(\pi (2n-1))}{2n-1}$$
for a rectangular channel flow, where the top half of the inlet flow has a concentration of 1, and the bottom half of the inlet flow has a concentration of 0. This infinite sum can be truncated after a desired accuracy is reached, and the problem can be solved explicitly.

#### 3.1.3. Droplet Microfluidics

For the last use case, the dynamics of the droplet in an immersed fluid is simulated. The droplet position is calculated based on the droplet speed $u_{d}$, i.e.,
(11)$$s=|u_{d}|t,$$
where the distance traveled over time *t* by the droplet is given by *s* and the droplet speed is determined according to
(12)$$u_{d}=\alpha \frac{Q}{A}.$$Here, *Q* is the volumetric flow rate of the channel, *A* is the area of the channel cross-section, and α is the slip factor [38]. The path *s* will be taken through the middle of the channel and, in case of a corner, the droplet will be assumed to go around the corner with a constant velocity magnitude. Using this, the location of the droplet can be properly determined, but not the deformation of the droplet. A heuristic approach can be adopted to predict droplet stability, however, this is outside the scope of this case study.

### 3.2. Finite Volume Method

The first CFD method that will be discussed is the *Finite Volume Method* (FVM [10]). Together with the other conventional CFD methods (the Finite Difference and Finite Element Method), the FVM puts the focus on solving the Navier–Stokes equations [9]. The Navier–Stokes equations are the governing equations for fluid dynamics and are given by the mass equation
(13)$$\frac{\partial \rho }{\partial t}+\nabla \cdot \left(\rho u\right)=0$$
and the momentum equation
(14)$$\frac{\partial }{\partial t}\left(\rho u\right)+\nabla \cdot \left(\rho uu^{T}\right)=-\nabla p+\nabla \cdot \sigma ,$$
where the effect of gravity is neglected. Here, ρ is the density, u is the velocity vector, *p* is the pressure field, and σ is the stress tensor. Together with an equation of state, the Navier–Stokes equations create a system of equations that can be solved. The domain is discretized into a set of cells, and each unknown is solved for and averaged in each cell. The flux on the boundary between neighboring cells can be solved using Riemann solvers. This scheme is conservative by construction, but is inherent to numerical diffusion for low-order schemes [10,14]. The FVM approach was implemented in the tool OpenFOAM [39], an open-source FVM solver. More information on the FVM can be found in dedicated works by [9,10].

### 3.3. Lattice-Boltzmann Method

The *Lattice-Boltzmann Method* (LBM, [11]) was derived from the method of *Lattice Gas Automata* (LGA, [40]). This method discretizes the Boltzmann equation, rather than the Navier–Stokes equations, but it can be proven to solve the Navier–Stokes equations on the macro-scale using the Chapman–Enskog theory [41,42]. The Boltzmann equation is given by
(15)$$\frac{\partial f}{\partial t}+\xi_{\beta }\frac{\partial f}{\partial x_{\beta }}+F_{\beta }\frac{\partial f}{\partial \xi_{\beta }}=\Omega \left(f\right).$$Here, the Einstein notation for summation is used and f(x,ξ,t) is the particle distribution function, which is a fundamental variable in kinetic theory that depends on particle location x and velocity ξ. *F* is a force field acting on the particles and Ω(f) is the collision operator which describes the behavior of particles in the event of a collision, i.e., when particles collide with each other.

In the LBM, the velocity space of the particle distribution function *f* is discretized. This discretization happens in a certain dimensionality on so-called lattices. An example of an LBM lattice is the D2Q9 lattice, which is a two-dimensional lattice on the square [−1,1]×[−1,1] in the Cartesian plane with a set of nine discrete velocities. A graphical representation of the D2Q9 lattice is given in Figure 5a, where the discrete velocities are numbered 0 through 8 and the 0th velocity is called the rest velocity. Analogously, an example of a three-dimensional lattice on the cube [−1,1]×[−1,1]×[−1,1] in the Cartesian coordinate system with 19 discrete velocities is given in Figure 5b. The lattices are chosen in this way, because they are mathematically compatible to solve the Navier–Stokes equations on the macroscopic level.

The operations performed on these lattices are the collision and streaming of particle densities. In this case study, the *Bhatnagar–Gross–Krook* (BGK) collision operator was considered, i.e.,
(16)$$\Omega_{i}\left(f\right)=-\frac{f_{i}-f_{i}^{\mathrm{eq}}}{\tau }\Delta t.$$Here, τ is the relaxation time, and $f^{\mathrm{eq}}$ is the particle distribution function of the fluid when it is in equilibrium. It is given as
(17)$$f_{i}^{\mathrm{eq}}\left(x,t\right)=w_{i}\rho \left(1+\frac{u\cdot c_{i}}{c_{s}^{2}}+\frac{{\left(u\cdot c_{i}\right)}^{2}}{2c_{s}^{4}}-\frac{u\cdot u}{2c_{s}^{2}}\right).$$The complete operation, i.e., the collision and streaming of particle densities, during each timestep is defined as
(18)$$f_{i}\left(x+c_{i}\Delta t,t+\Delta t\right)=f_{i}\left(x,t\right)\left(1-\frac{\Delta t}{\tau }\right)+f_{i}^{\mathrm{eq}}\left(x,t\right)\frac{\Delta t}{\tau }.$$After the collision operator, the particle densities are streamed to the neighboring lattices, and the process starts over again. The LBM approach was implemented using the tool Palabos [43], an open-source LBM solver. The numbering of the velocities in Figure 5 is consistent with the velocity numbering in Palabos. For a more detailed explanation of the LBM, the reader is referred to a more dedicated work by [12].

## 4. Conducted Simulations

Using the approaches reviewed above, all use cases introduced in Section 2 have been simulated. In this section, we describe the corresponding instances as applied in this case study. More precisely, we summarize the precise setup of the use cases (e.g., the geometry of the considered channels, the characteristics of the used fluids, boundary and initial conditions, etc.) as well as the setup of the corresponding simulation approaches (e.g., the discretization of the geometry, steady-state or transient simulation, etc.). Based on that, all simulations have been conducted using a single thread on an *AMD Ryzen 9 3950X 16-Core* processor with 128 GB *DDR4* RAM. As solvers, we used in-house Python scripts for the 1D approach, OpenFoam for the FVM, and Palabos for the LBM (the latter two for both the 2D and 3D approaches). The respective source code has been published and can be found in our GitHub repository *Microfluidics-Abstraction-Levels* [44]. All simulations were run without any explicit parallelization to acquire a representative comparison. All obtained results are eventually presented and discussed in Section 5.

### 4.1. Non-Newtonian Fluid Flow

#### 4.1.1. Setup of the Use Case

For this use case, we considered a simple rectangular channel as shown in Figure 6a with length l=500 μm, width w=100 μm and, if three dimensions are considered, the height h=w. The initial condition is a stagnant flow with no-slip boundary condition at the walls. At the inlet, a uniform flow velocity $u_{\mathrm{in}}$ of $10\frac{mm}{s}$ has been applied, and a zero-gradient Neumann boundary condition was set for the pressure. At the outlet, a Dirichlet pressure condition is applied, $p_{\mathrm{out}}=0$, and a zero-gradient Neumann boundary condition was set for the velocity. The flow profile is taken at the measurement line *m*, 400 μm downstream. We used the parameters of blood for this use case, i.e., the density was set to 1060 $\frac{g}{m^{3}}$ [45] and the viscosity parameters for the Carreau–Yasuda models were set according to the values provided in Section 2.2.

#### 4.1.2. Setup of the Simulation

For the 1D approach, a simple Python script was created to solve the local viscosity from the Carreau–Yasuda model along the *y*-axis of the channel. The Newton–Rhapson method was used as root-finding method to find the viscosity and the Simpson’s rule was implemented as quadrature rule to determine the local velocity from Equation (Equation 7).

To realize the FVM, we used the steady state solver *simpleFoam* in OpenFOAM, with relaxation factors 0.3 and 0.7 for the pressure and velocity fields, respectively. The domain has been discretized with a Cartesian grid of 500 by 100 cells for the 2D approach, and 500 by 100 by 100 cells for the 3D approach. Iterative solvers were used to solve the pressure and velocity fields to a tolerance of 1 × 10^−8^. To get the shear strain rate $\dot{\gamma }$ in the FVM, the gradient of the velocity field is solved for explicitly every iteration step.

For the LBM, the discretization has been done in the same fashion as for the FVM. The D2Q9 and D3Q19 lattices were used for the 2D and 3D approaches, respectively. The LBM is an intrinsic transient method, so in order to get the steady-state solution, Palabos solved the flow field until a convergence was reached. The convergence tolerance was set to 1 × 10^−5^ for both solvers. In the LBM, the shear strain rate is available locally and no finite difference scheme is necessary [46].

### 4.2. Fluid Mixing

#### 4.2.1. Setup of the Use Case

The geometry of the domain for the mixing simulation is a straight channel as shown in Figure 6b with one input for the two different fluids, and one outflow. The length *l* of the channel is set to 600 μm, the width w=40 μm and, if three dimensions are considered, the height h=w. A no-slip boundary condition is set at the walls. At the inlet, a Dirichlet velocity $u_{\mathrm{in}}=10\frac{mm}{s}$ and Dirichlet concentration $C_{\mathrm{in}}$ boundary condition is set. The concentration boundary condition is set to $C_{1}=1$ for the left fluid, and $C_{2}=0$ for the right fluid. At the outlet, a Dirichlet pressure boundary condition $p_{\mathrm{out}}=0\;Pa$ is set. At the initial condition, the fluid is stagnant, and the concentration is set to $C_{1}$ and $C_{2}$ for the left half and right half of the geometry, respectively. The resulting concentration distribution is taken at measurement line *m*, 500 μm downstream. The density of both fluids A and B was set to 1 $\frac{kg}{m^{3}}$ for simplicity, and the diffusivity parameter $D_{12}$ of the two fluids was set to 90 $\frac{\mu m^{2}}{s}$.

#### 4.2.2. Setup of the Simulation

For the 1D approach, a Python script was written to implement Equation (Equation 10), which could be solved explicitly. The sum can be truncated as every new sum becomes rapidly negligible, and it was truncated with an error term of $O(10^{-12})$.

For the FVM, the *Volume-of-Fluids* (VOF, [47]) method was used to solve the multi-component system. The problem was solved using OpenFOAM’s transient solver *twoLiquidMixingFoam* until the concentration field converged. The geometry was discretized using a Cartesian grid of 600 by 40 cells for the 2D approach, and 600 by 40 by 40 cells for the 3D approach.

For the LBM, the Dirichlet boundary condition was set using Bounceback dynamics at the walls, and Bounceback dynamics with adapted wall velocity were set at the inlet and outlet to get the Dirichlet boundary conditions at the open boundaries. The domain was discretized with the same Cartesian grid as for the FVM. Following the approach from [48], the Shan–Chen method [49] was used to solve the multi-component problem. This method adds an artificial force to the fluid elements that mimics the separation of different components. The Shan–Chen force is controlled with a coefficient *G*, which controls the strength of the interaction between different species [12]. The value depends on the size of the cells in the grid. For the mixing problem, this value controls the diffusivity and for this use case, it was carefully set to 0.8.

### 4.3. Droplet Microfluidics

#### 4.3.1. Problem Setup

The geometry of the domain is the bifurcation given in Figure 6c with the length l=500 μm, the length k=250 μm, the width w=100 μm, and, if three dimensions are considered, the height h=100 μm. A no-slip boundary condition is set at the walls. At the inlet, a Dirichlet velocity boundary condition of $u_{\mathrm{in}}=10\frac{mm}{s}$ is set. At the two outlets, Dirichlet pressure boundary conditions are set. These are $p_{\mathrm{out}}^{1}=10$ Pa, and $p_{\mathrm{out}}^{1}=0$ Pa to ensure that the droplet will follow path *s* into the top channel. The initial velocity in the domain is 0, and the droplet is sized such that it practically fills the entire cross-section of the channel (a layer of cells filled with carrier fluid is present between the droplet and the walls to assure stability). The droplet length $d_{l}$ is 200 μm. The density of the carrier fluid and the droplet was set to $1\;\frac{kg}{m^{3}}$, the viscosity of both fluids to $1.004\;\frac{\mu m^{2}}{s}$, and the surface tension is set to $\gamma =0.005\;\frac{N}{m}$.

#### 4.3.2. Implementation

For the 1D approach, the droplet is assumed to flow along path *s* from Figure 6c with a speed of $u_{d}$, which depends on the volumetric flow rate. The hydraulic resistances of the channels were calculated through Equations (Equation 2) and (Equation 5), and the corresponding volumetric flow velocities are $5.07\;\frac{m}{s}$ and $4.93\;\frac{m}{s}$ for the top and bottom channels, respectively, for a slip factor α of 1.28 [50]. The position of the tailing edge of the droplet was used to determine the location of the droplet. Assuming no droplet deformation, the remaining length $d_{l}$ of the droplet was mapped in front of this point.

For the FVM, OpenFOAM’s transient solver *interFoam* was used. It takes only a few timesteps to resolve the internal velocity field, which does not affect the shape of the droplet. As with the mixing problem, the VOF method was used to solve the multi-component problem. The geometry was discretized using a Cartesian grid, where each cell had sides of 1 μm.

For the LBM on the other hand, the problem was first simulated without a droplet, to acquire a converged velocity field, which was subsequently used as the initial condition for the velocity field of the droplet deformation problem. This was done, because initializing the velocity at 0 causes a set of artificial pressure waves through the domain until the pressure and velocity field are in equilibrium. This effect destroys the droplet and effectively renders the simulation meaningless. The velocity boundary conditions on the walls, inlet, and outlets were set using Bounceback dynamics. The pressure Dirichlet boundary condition was set using Antibounceback dynamics. For the LBM, the same Cartesian grid was used as for the FVM. The Shan–Chen method was used to solve this multi-component problem. Here, the coefficient *G* controls the numerical surface tension of the droplet and it depends on the size of the cells in the grid. It was carefully chosen at 1.1.

## 5. Obtained Results and Discussion

Eventually, this section summarizes all results obtained by the case study described above and draws corresponding conclusions from them. Following the structure of all sections above, each case is discussed separately in the following.

### 5.1. Non-Newtonian Fluid Flow

The resulting flow profiles are probably the most relevant results generated by the simulators for this use case. Correspondingly obtained results are given in Figure 7 for all simulation approaches. The flow profile is taken along the width of the channel at the measurement line *m* in Figure 6a. On the *x*-axis, the velocity of the stream is given in $\frac{mm}{s}$ and, on the *y*-axis, the position along the width of the channel is given in μm. The correspondingly needed runtimes in CPU seconds and required memory in MB are summarized in Table 1. For the 1D approach, no runtimes/memory requirements are reported since all simulations can be completed in negligible time (i.e., less than a second).

The plots clearly show that the results from the 1D and the 2D approaches are rather close to each other—indicating a similar quality for both of them. Considering that the 1D approach requires almost no computational efforts (while the 2D approaches may take some minutes), this provides a clear indication to choose the former over the latter for straight channel use cases. If 3D is considered, different results are obtained, i.e., the maximum velocity of the 3D simulations is significantly higher than that of the 2D simulations. This is to be expected, as the inlet flow speed is set equal for all simulations, and the 3D simulations have the wall effect of four walls instead of only two walls—resulting in a higher maximum velocity, because the fluid is being “squeezed” through a tinier gap. However, this higher precision comes at a larger computational cost: In case of FVM, the runtime even increases by a factor of almost 165; in case of LBM by a factor of about 71.

Overall, this allows for the conclusions that, if a rough approximation is sufficient for the end-user’s needs, the 1D simulation certainly delivers that at almost no computational costs. If instead very precise results are needed, the end-user probably should take the extra mile and use the 3D simulation (despite its computational costs). In this example, the 2D simulations hardly provide any further benefits compared to the 1D approach, however, one should keep in mind that the 2D approach can also be used for more complex two-dimensional flow problems. There is no significant difference in the computational cost, between the FVM and LBM approaches, and both give adequate results for this use case.

### 5.2. Fluid Mixing

The mixing performances are probably the most relevant results generated by the simulators for this use case. This is quantified using the *Absolute Mixing Index* (AMI) as given by [51], i.e.,
(19)$$\mathrm{AMI}=\frac{\sqrt{\frac{1}{N}\sum_{i=1}^{N}{\left(\alpha_{i}-\langle \alpha \rangle \right)}^{2}}}{\langle \alpha \rangle },$$
where α represents the concentration of fluid A coming in on the left, normalized on a scale from 0 to 1. The average value of α over all *N* data points is given by 〈α〉.

The results for the mixing use case taken at cross-section *m* in Figure 6b are summarized in Figure 8 and Figure 9. The results of the 1D and 2D approaches are shown in Figure 8, where the *x*-axis denotes the position along the channel width and the *y*-axis denotes the concentration of fluid A. The concentration of fluid A as obtained by the 3D FVM and 3D LBM approaches is given in Figure 9a,b, respectively. Here, completely red means a 100% concentration of fluid A and completely blue means a 100% concentration of fluid B. The correspondingly acquired AMI values, needed runtimes, and memory requirements are provided in Table 2. Again, for the 1D approach, no runtimes are reported since all simulations can be completed in negligible time (i.e., less than a second).

For this use case, the simulation results are much more inconclusive. In fact, there obviously is a huge difference between the results obtained by the FVM method and the LBM method. Since we considered a straight channel during the mixing, the 1D simulation (relying on an analytical solution that is exact in this case) does provide a ground truth—verifying that the results obtained by FVM are correct, while the results obtained by LBM are far off. This does not mean that the LBM approach cannot be used for miscible flow simulation, but it does indicate that the Shan–Chen multiphase method is probably not the best method for this use case.

As for runtime performance, the 1D solution obviously is the best option again. Considering that, for this scenario, this even provides a ground truth, it obviously is the best choice in general. However, this only holds for straight channels. As soon as, e.g., a meandering channel is considered, the 1D approach is not exact anymore, and designers or researchers must rely on CFD methods (and need to accept the longer runtimes). Then, the previously reported results can be seen as an indication that, having the option between the described methods, FVM seems to lead to better results. However, a physical prototype might be needed to give a decisive answer to which approach indeed provides the best quality. Overall, these differences clearly show that just trusting a single simulation engine can easily be misleading and may serve the impression that simulations always give the end-users accurate results. Using different schemes, leading to different results, may increase awareness that simulations are not always perfect, and should be critically reflected upon.

### 5.3. Droplet Microfluidics

Finally, the positions and possible deformations of the droplets are probably the most relevant results generated by the simulators for this use case. Correspondingly obtained results are summarized in Figure 10 for the 1D, 2D, and 3D approaches. More precisely, for each case, the position and the shape of the droplet is shown (denoted as red entity) for three timestamps. Note that, since the 1D approach is incapable of simulating the shape of the droplet, its position is simply denoted in terms of a block-like entity in Figure 10a. The correspondingly needed runtimes and memory requirements are provided in Table 3. Here, no runtimes are reported for the 1D approach since all simulations can be completed in negligible time (i.e., less than a second).

The results for this use case perfectly show the trade-off between quality and computational effort. In fact, the results from the 1D approach (generated in almost no time), provide rather accurate positions (in fact, the position of the droplet is almost identical in 1D, 2D, and 3D). Hence, if the end-user is just interested in the positions of droplets, 1D certainly is the way to go. However, one still has to be careful for the slight difference that is observable between the 1D and 3D approaches. In fact, these differences might accumulate for more complex channel configurations.

If also deformation (or stability) of the droplet is considered, then the 1D approach is not suitable at all and the end-user has to opt for 2D, or even 3D, approaches. This obviously comes at a substantially higher computational cost, but it seems that a 2D approach already comes with an acceptable accuracy, i.e., it might not always be necessary to spend the substantially larger runtime required for 3D approaches. In fact, although the droplet seems to have a lower velocity in the 2D approaches, all 2D and 3D approaches agree that the droplet does not break. If a designer or researcher wishes to have more detailed information on the droplet deformation, care should again be taken, as it is clear that none of the approaches really agree on how the droplet deforms.

## 6. Conclusions

In this work, we considered the simulation of pressure-driven and channel-based microfluidic devices at different levels of abstraction. To this end, we considered three representative use cases, namely the non-Newtonian flow in a channel, the mixing of two fluids in a channel, and the behavior of droplets in channels. The case study and the obtained results clearly show that simulations are certainly not perfect but frequently provide insights that can aid and improve the design process. At the same time, the case study shows that the broad spectrum of simulation approaches leads to different performances with respect to accuracy and required compute time.

Table 4 provides a color-coded overview of the correspondingly obtained “take-home messages” for each use case (columns) and abstraction level (rows). Here, green denotes the best possible option (e.g., simulating a non-Newtonian fluid flow in a rectangular channel is best conducted using the 1D abstraction as it provides precise results in negligible runtime), while red denotes an option which should be avoided (e.g., simulating a fluid mixing process within a straight channel in 2D or 3D takes too much time, and does not yield much extra information in return) or which is not applicable at all (e.g., splitting and deformations in droplet microfluidics cannot be simulated at all in 1D). Orange denotes options which could be useful, but usually involve a trade-off (e.g., simulating deformations in droplet microfluidics in 2D may not provide perfect accuracy but, considering the much less runtime compared to 3D, may provide an acceptable compromise).

Overall, this clearly shows that designers and researchers should be aware of the available solutions and accordingly trade-off these performances with their respective needs. With this, we believe that this work provides a contribution towards increasing the awareness and the understanding of the potential of simulation for microfluidic devices.

Future work certainly should focus on improving the accuracy and reliability of the simulation results—particularly towards avoiding contradictory or inconclusive results. This could be accounted for through the introduction of an error margin that allows for a bigger solution space—increasing the chance that the ground-truth is considered. This error margin could lead designers towards more robust designs of microfluidic devices. Additionally, a better understanding about computational efforts vs. quality of simulations at the different abstraction levels (as fostered through this work) may pave the way towards hybrid solutions, i.e., a combination of simulation approaches where the overall design is simulated at high abstractions (fast, but less accurate) and corner case components are simulated at lower abstractions (slower, but for smaller parts and with better accuracy). For all these endeavors, the insights gained through this case study provide a good basis.

## Figures and Tables

**Figure 1 sensors-22-05392-f001:**
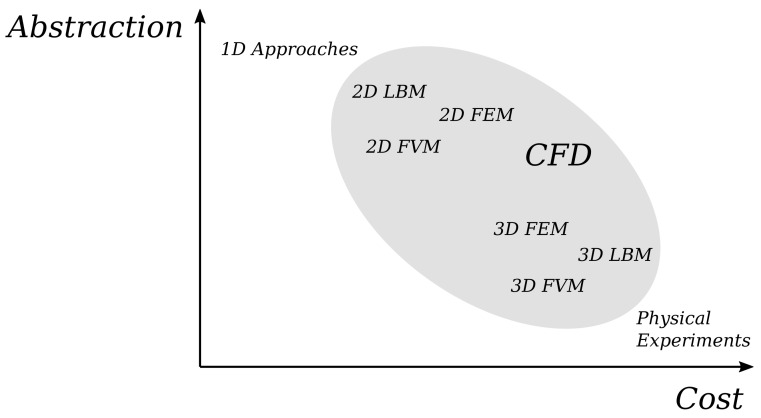
Abstraction levels for simulating microfluidic flow.

**Figure 2 sensors-22-05392-f002:**
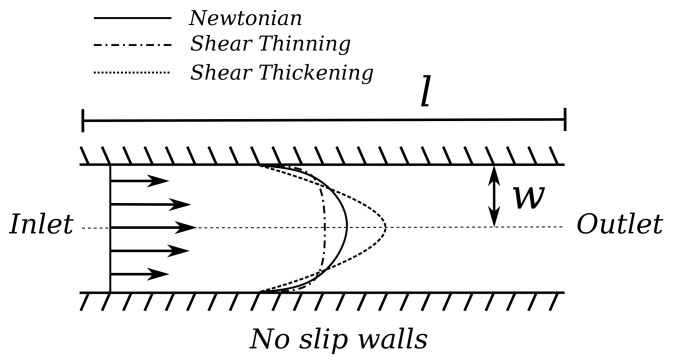
A set of flow profiles through a channel of length *l* with width *w*, where the walls on the top and bottom are assumed to be rigid. The flow profile of a Newtonian fluid is shown by the solid line and the flow profiles for the shear thinning and thickening fluids are given by the dashed lines.

**Figure 3 sensors-22-05392-f003:**
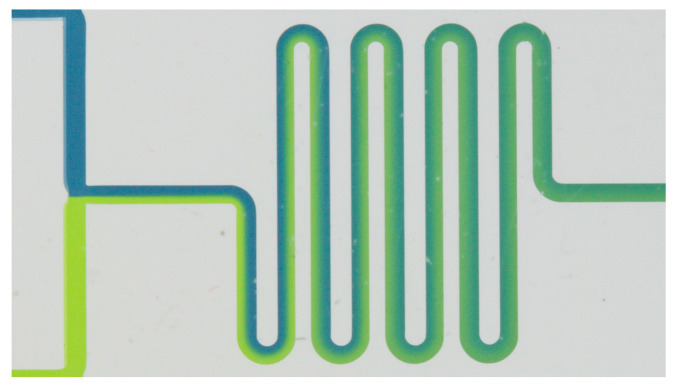
Flow mixing in a meandering channel [28].

**Figure 4 sensors-22-05392-f004:**
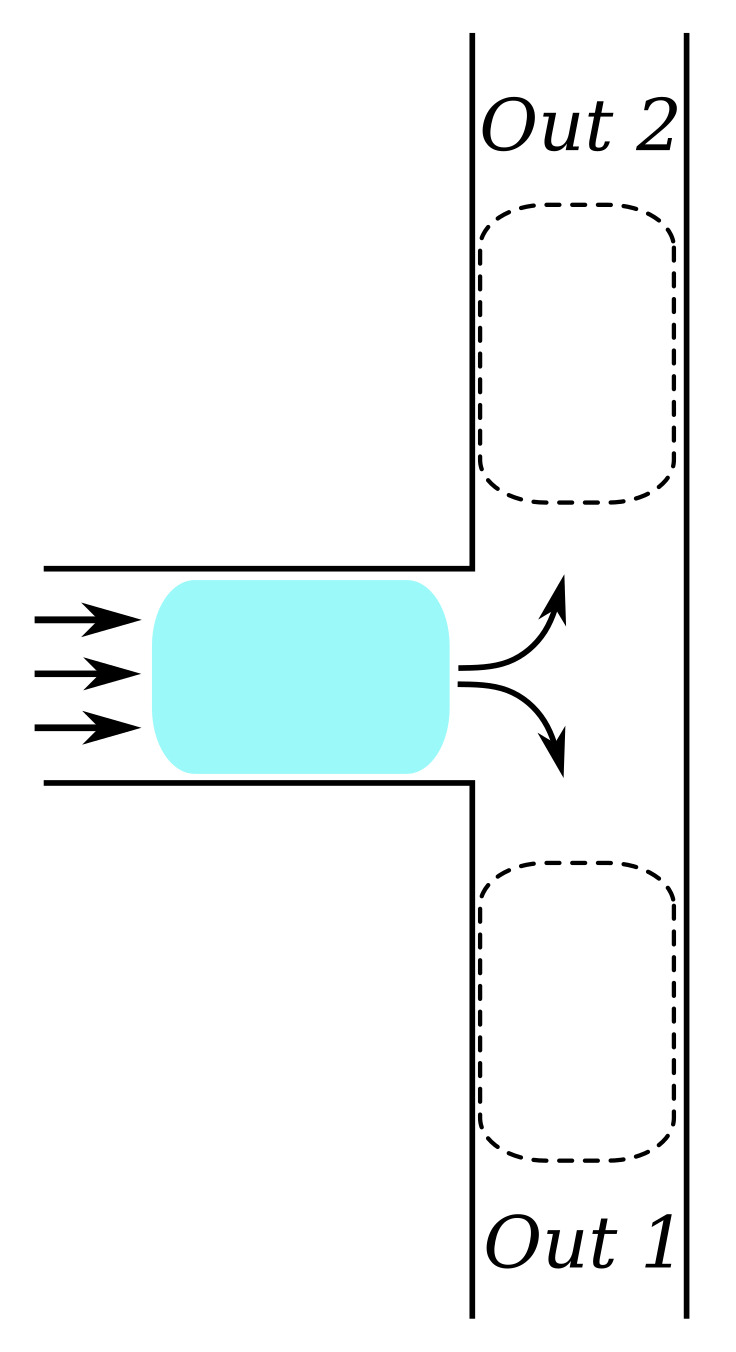
Microfluidic channel junction with droplet immersed in carrier fluid.

**Figure 5 sensors-22-05392-f005:**
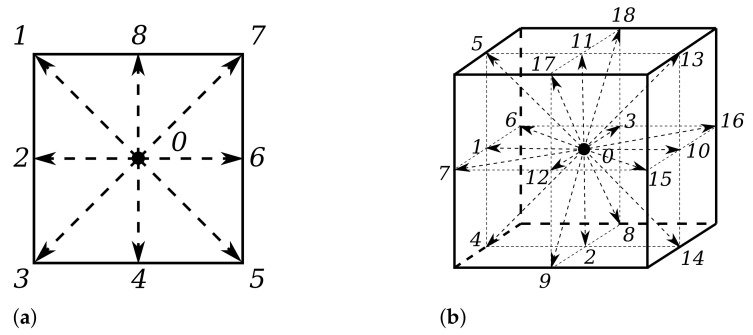
The lattices that are used for the LBM simulations. (**a**) The D2Q9 lattice for 2D LBM simulations; (**b**) The D3Q19 lattice for 3D LBM simulations.

**Figure 6 sensors-22-05392-f006:**
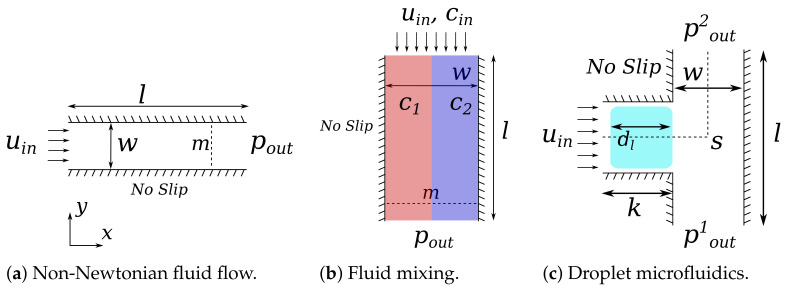
Geometries of the simulation domains.

**Figure 7 sensors-22-05392-f007:**
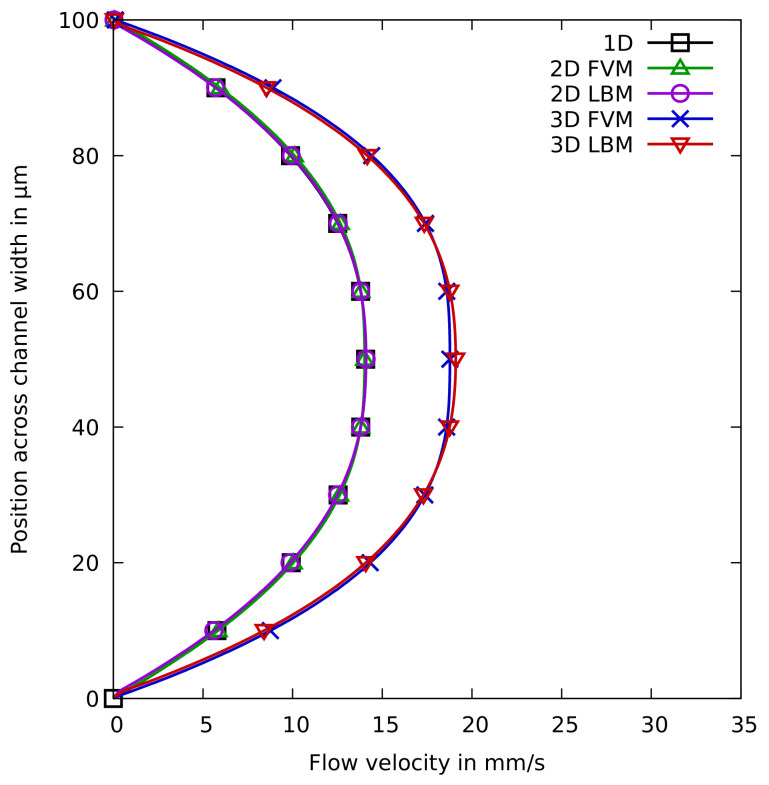
Results of the simulations for the non-Newtonian fluid flow use case.

**Figure 8 sensors-22-05392-f008:**
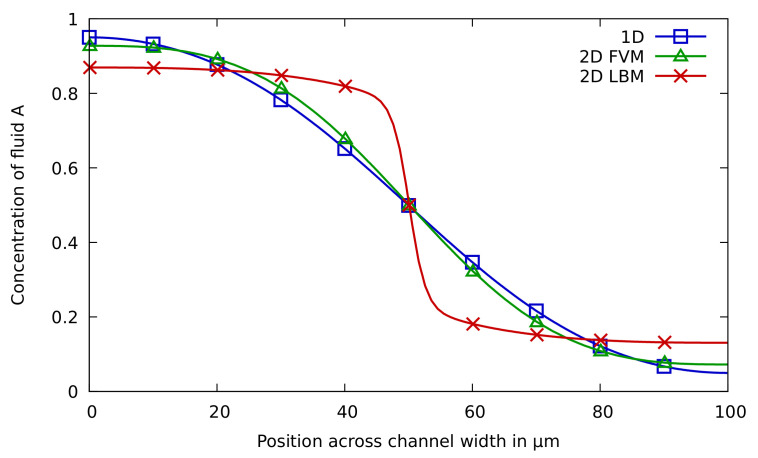
The concentration distribution of fluid A for the 1D approach, the 2D FVM approach, and the 2D LBM approach. For the 2D approach, the concentration distribution was taken at the measurement line *m* in Figure 6b.

**Figure 9 sensors-22-05392-f009:**
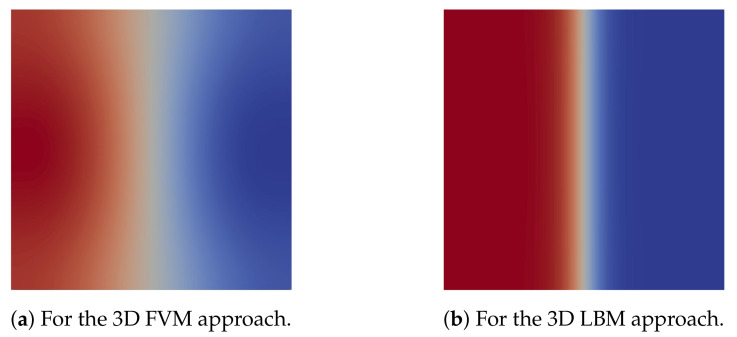
Cross-sectional concentration distribution of fluid A at measurement line *m* in Figure 6b.

**Figure 10 sensors-22-05392-f010:**
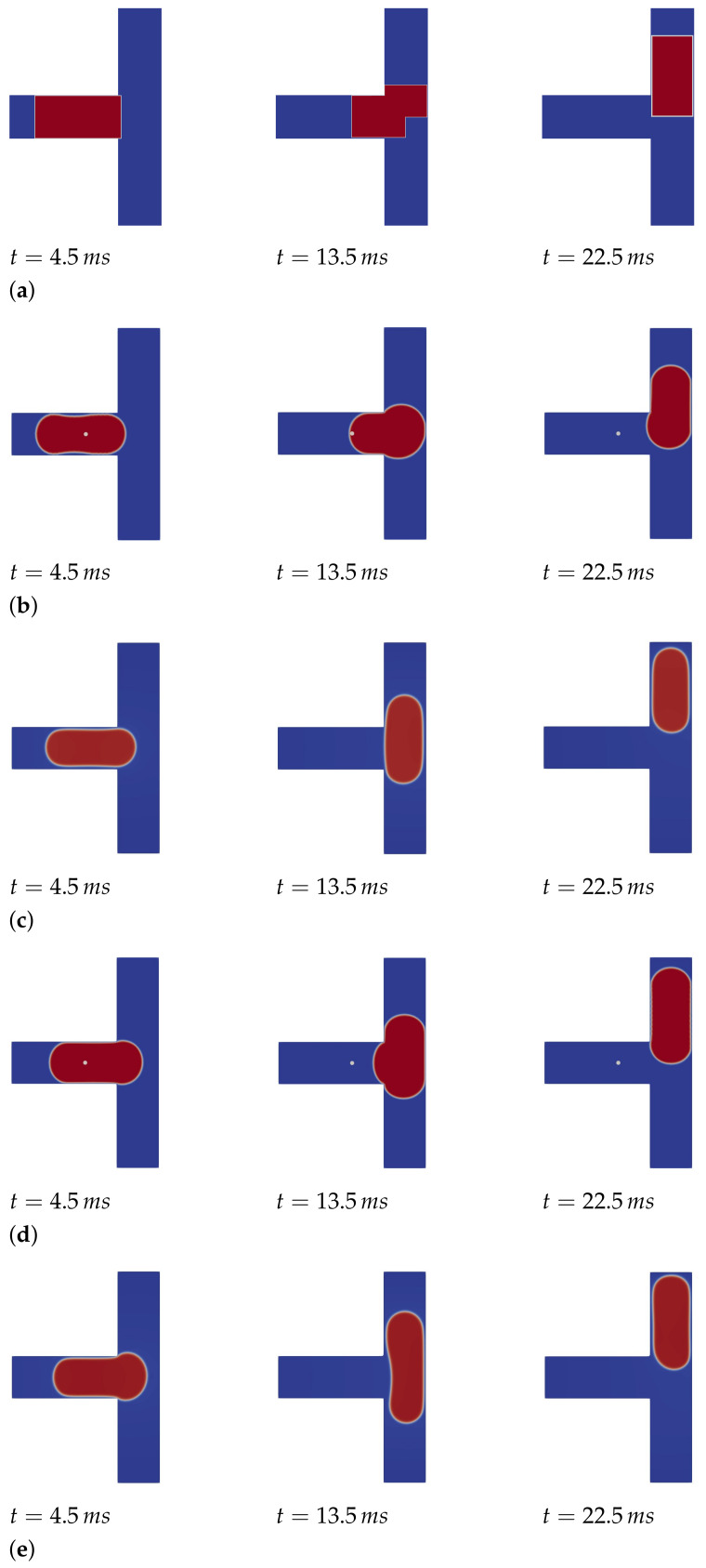
Results of the simulations for droplet microfluidics. (**a**) Droplet position at different timestamps as simulated with the 1D approach; (**b**) Droplet position and deformation at different timestamps as simulated with the 2D FVM approach; (**c**) Droplet position and deformation at different timestamps as simulated with the 2D LBM approach; (**d**) Droplet position and deformation at different timestamps as simulated with the 3D FVM approach. (**e**) Droplet position and deformation at different timestamps as simulated with the 3D LBM approach.

**Table 1 sensors-22-05392-t001:** The CPU time and required memory in MB for the 2D and 3D simulations.

	2D FVM	2D LBM	3D FVM	3D LBM
Time [hh:mm:ss]	00:02:34	00:05:31	07:02:27	06:31:32
Required Memory [MB]	132.8	14.6	5819.1	1410.1

**Table 2 sensors-22-05392-t002:** Absolute Mixing Index (AMI) for all mixing simulations, the runtime in CPU time, and required memory in MB for the 2D and 3D simulations.

	1D	2D FVM	2D LBM	3D FVM	3D LBM
AMI	0.654	0.665	0.680	0.654	0.519
Time [hh:mm:ss]	-	00:06:38	00:08:18	11:44:36	13:52:23
Required Memory [MB]	-	109.6	20.0	1530.0	515.1

**Table 3 sensors-22-05392-t003:** Runtime in sec and required memory space for the 2D and 3D approaches.

	2D FVM	2D LBM	3D FVM	3D LBM
Time [hh:mm:ss]	00:03:29	00:02:23	06:06:13	04:26:45
Required Memory [MB]	96.1	32.4	898.1	711.6

**Table 4 sensors-22-05392-t004:** A color-coded overview of the performance of all approaches for all use cases.

Level	Non-Newtonian Fluid Flow		Fluid Mixing		Droplet Microfluidics		
	Rectangular Channel	Other	Straight Channel	Meander	Position	Split	Deformation
**1D**							
**2D**			*	*			
**3D**			*	*			

* The Shan–Chen approach is probably not the best fit for LBM.

## Data Availability

Not applicable.

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
