# Peer review of "Simulation of Pressure-Driven and Channel-Based Microfluidics on Different Abstract Levels: A Case Study"

_sensors, 2022, doi:10.3390/s22145392_

Round 1

Reviewer 1 Report

The authors reviewed, compared and discussed, regarding advantages and limitations, different methods of fluid simulation for lab-on-a-chip applications. They considered, on one side, simulation approaches in 1D, which are based on analytical solutions of simplified problems, and approaches in 2D and 3D, using Finite Volume Method and Lattice-Boltzmann Method. For the comparison, the authors approached three case studies: non-Newtonian flow in a channel, mixing of two fluids in a channel, and the behavior of droplets in channels.

Although none of the referred methods are novel (all of them are well known in literature), their comparison may be relevant for researchers working in the area, which brings some merit to the research. I have a few remarks and comments for improving the manuscript:

-        While the simulation methods are compared between them, no experimental validation in given (for the results presented in figures 7, 8, 9 and 10). The paper would benefit from an experimental validation of the results in similar conditions as the simulated ones. That additional curve in the plots would be an additional validation step for the different abstraction levels.

-        Even using the same method, the simulation results may differ. For instance, for FVM, the discretization method, the quality of the mesh, number of elements or even the quality of the selected solver will affect the accuracy of the results. This needs to be addressed in the manuscript.

-        Following the discussion, the authors should add a table summarizing the advantages and disadvantages of all methods.

-        Please check if the authors have authorization for reprinting the figures from literature.

Reviewer 2 Report

Takken and Wille presented a case study of numerical analysis of pressure driven and channel-based microfluidics by using 1D, 2D and 3D approaches. In this work, the authors implemented computational Fluid Dynamics (CFD) methods including the Finite Volume Method (FVM) and the Lattice-Boltzmann Method (LBM) for the 2D and 3D analyses, and applied analytical solutions of simplified problems for the 1D case.  For the numerical work, non-Newtonian flow, the mixing of two fluids, and the behavior of droplets in channels are considered. In the first example, shear thinning and shear thickening fluids are considered. For the fluid mixing, a passive mixing example is simulated where the mass transport happens through diffusion governed by Fick's law. Finally, for droplet microfluidics, a T-junction is considered that deals with droplet motion at the bifurcation of the channel.  Overall, this work a good comparison of simulation tools applied in microfluidics at different levels of abstraction. The manuscript has a good flow and is well-structured. Although the novelty of the work is not very high, the information provided in the manuscript can be useful especially for the early career researchers and graduate students. I have the following specific comments:

-A discussion of the viscocity and fluid mixing could be useful. 

-In figure 9a, there seems to be slight variation of white intensity between the top and bottom of the area. Why? 

Reviewer 3 Report

In this manuscript, the authors presented a thorough overview of the major methods used for the simulation of microfluidics. The writing is very concise and clear, with good illustrations and formula formatting. This topic could be practically useful for microfluidic researchers to choose the proper simulation method for their specific needs and budgets. I have no comments on the major part of this manuscript, however, the inclusion of some real experimental results would be better to prove the usefulness of these simulation methods overall.  

Round 2

Reviewer 1 Report

The authors satisfactorily answered my questions and concerns. Experimental validation would be valuable but, given the said impossibility to perform the experimental analysis, the manuscript can be accepted as it is.